# Topical Antibiotic Therapy in the Ocular Environment: The Benefits of Using Moxifloxacin Eyedrops

**DOI:** 10.3390/microorganisms12040649

**Published:** 2024-03-25

**Authors:** Lorenzo Drago

**Affiliations:** 1UOC Laboratory of Clinical Medicine with Specialized Areas, IRCCS Multimedica Hospital, 20138 Milan, Italy; lorenzo.drago@unimi.it; 2Clinical Microbiology and Microbiome Laboratory, Department of Biomedical Sciences for Health, University of Milan, Via Mangiagalli 31, 20133 Milan, Italy

**Keywords:** quinolones, moxifloxacin, ocular infections, ocular microbiota, prophylaxis

## Abstract

Moxifloxacin is a fourth-generation fluoroquinolone antibiotic available for ophthalmic use. It inhibits two enzymes involved in bacterial DNA synthesis, covering Gram-positive and Gram-negative pathogens. This spectrum allows for the formulation of self-preserving bottle solutions, while its interesting pharmacological profile is distinguished by efficacy at low tissue concentrations and by an infrequent dose regimen due to its long duration on ocular tissues. This enhances patient compliance, promoting its use in children. The human eye hosts several microorganisms; this collection is called the ocular microbiota, which protects the ocular surface, assuring homeostasis. When choosing an antibiotic, it is appropriate to consider its influence on microbiota. A short dose regimen is preferred to minimize the impact of the drug. Moxifloxacin eyedrops represent an effective and safe tool to manage and prevent ocular infections. As healthcare providers face the complexity of the ocular microbiota and microbial resistance daily, the informed use of moxifloxacin is necessary to preserve its efficacy in the future. In this regard, it is well known that moxifloxacin has a lower capacity to induce resistance (an optimal WPC and MPC) compared to other quinolones, but much still needs to be explored regarding the impact that fluoroquinolones could have on the ocular microbiota.

## 1. Introduction

Ocular infections and related disorders continue to be a significant public health concern, affecting millions of individuals worldwide [1]. The delicate nature of the eye demands precise and targeted treatment strategies to ensure the effective resolution of infections while minimizing adverse effects. In recent years, moxifloxacin has emerged as a prominent therapeutic option in the form of eye drops, demonstrating efficacy in treating a broad spectrum of ocular infections [2].

Moxifloxacin is a fourth-generation fluoroquinolone antibiotic that exhibits broad-spectrum activity against both Gram-positive and Gram-negative bacteria. Its unique chemical structure, which includes an eight-methoxy group, enhances its potency and efficacy. Moxifloxacin exerts its bactericidal effects by inhibiting DNA gyrase and topoisomerase IV enzymes, critical components in bacterial DNA replication and repair processes [3]. This dual mechanism of action contributes to its effectiveness against a wide range of ocular pathogens, including *Staphylococcus aureus*, *Streptococcus pneumoniae*, *Haemophilus influenzae*, and *Pseudomonas aeruginosa*.

The ocular administration of moxifloxacin is a pivotal aspect of its use in ophthalmology. When applied topically as eye drops, moxifloxacin demonstrates high bioavailability on the ocular surface, such as in the aqueous and vitreous humors, leading to effective therapeutic concentrations at the site of most frequent ocular infections (such as conjunctivitis). The rapid diffusion of moxifloxacin into ocular tissues is attributed to its lipophilic and hydrophilic nature and its low molecular weight. These features allow the molecule to spread in the tear film and conjunctiva and diffuse efficiently through the cornea and other ocular barriers [4].

Furthermore, moxifloxacin exhibits prolonged retention in ocular tissues, enabling extended intervals between doses. This characteristic is particularly advantageous in enhancing patient compliance and in reducing the burden of frequent administration. The sustained therapeutic levels achieved with moxifloxacin contribute to its efficacy in treating both acute and chronic ocular infections [5].

The clinical applications of moxifloxacin in ophthalmology are several, encompassing a range of ocular conditions. Moxifloxacin eye drops are useful in the management of bacterial conjunctivitis and bacterial keratitis and as postoperative prophylaxis following ocular surgeries such as cataract extraction. The efficacy of moxifloxacin extends to complex cases, including those involving multidrug-resistant strains, making it a valuable tool in the armamentarium of ophthalmic therapeutics [6,7].

In addition to its antimicrobial properties, moxifloxacin exhibits anti-inflammatory effects, further contributing to its usefulness in ocular conditions with an inflammatory component. This dual action makes moxifloxacin a versatile option for practitioners dealing with a spectrum of ocular infections [8].

The use of moxifloxacin as eye drops can represent a significant advancement in the field of ophthalmic therapeutics. Its broad spectrum of activity, favorable pharmacokinetics, and clinical efficacy—particularly in children—make it a valuable tool in the management of various ocular infections [9,10,11]. The evolving landscape of moxifloxacin’s applications in ophthalmology underscores its continued relevance and contribution to the well-being of patients with ocular disorders.

Unlike other reviews, this paper aims to provide an updated overview of the general characteristics of moxifloxacin and its pharmacological profile, focusing on its employment in ophthalmology by looking inside the ocular microbiota and examining the homeostasis of the ocular surface.

## 2. Ocular Homeostasis: Safeguarding the Eye from Microbial Challenges

The human eye is a remarkable organ which is constantly exposed to the external environment and susceptible to several microbes, including potentially pathogenic ones. Remarkably, despite this continuous exposure, the eye maintains a state of homeostasis, minimizing the risk of infections. Ocular homeostasis involves a sophisticated interplay of various factors, including the integrity of ocular barriers, the content of tear antimicrobial proteins, the presence of saprophytic bacterial components, and the vigilant role of innate immunity [12].

The first line of defense in ocular homeostasis is the integrity of ocular barriers, primarily the cornea and the conjunctiva. The cornea, with its multilayered structure and tight junctions between epithelial cells, acts as a physical barrier, preventing the penetration of pathogens into the eye. Similarly, the conjunctiva, a mucous membrane covering the outer surface of the eye and the inner surface of the eyelids, secretes mucus and harbors specialized cells that contribute to the prevention of microbial invasion [13].

The presence of an intact blood–ocular barrier, formed by the tight junctions of the vascular endothelium within the eye, further limits the access of pathogens to intraocular structures. The coordinated function of these barriers serves as a crucial defense mechanism, protecting the delicate structures of the eye from potential threats.

Tears play a pivotal role in ocular defense, not only through lubricating the ocular surface but also by containing a rich array of antimicrobial proteins [14]. Lysozyme, lactoferrin, and secretory IgA are among the key components of tears with antimicrobial properties. Lysozyme, an enzyme that breaks down bacterial cell walls, acts as a potent bactericidal agent. Lactoferrin, an iron-binding protein, inhibits microbial growth by sequestering essential iron, and when instilled as eye drops in healthy eyes, it seems to act as a selective natural antiseptic by limiting pathogen growth while favoring saprophytic growth [15]. Secretory IgA provides immune protection by neutralizing and preventing the adherence of microbes to the ocular surface [16].

The dynamic composition of the tear film, which is constantly replaced by the lacrimal apparatus, creates an inhospitable environment for microbial colonization and plays the role of an essential component in maintaining ocular homeostasis.

Innate immunity also serves as a rapid and nonspecific defense mechanism against microbial challenges. The ocular surface is equipped with a variety of innate immune cells, including epithelial cells, dendritic cells, and macrophages. These cells actively participate in recognizing and eliminating pathogens through phagocytosis, the production of antimicrobial peptides, and the release of inflammatory mediators.

The complement system, a crucial component of innate immunity, is also present in tears, providing an additional layer of defense against microbial invaders. The coordinated action of these innate immune components contributes to overall protection against potential threats to ocular health [17].

In conclusion, ocular homeostasis is a complex and finely regulated process that involves the integration of various factors to protect the eye from microbial challenges. The integrity of ocular barriers, the content of tear antimicrobial proteins, the presence of saprophytic bacterial components, and the vigilant role of innate immunity collectively contribute to the maintenance of ocular health and the prevention of infections.

## 3. The Ocular Microbiome: Unveiling the Diversity and Dynamics of Ocular Microorganisms

The human eye, once considered a sterile environment, is now recognized as a dynamic ecosystem hosting a diverse array of microorganisms collectively known as the ocular microbiome. This emerging field of research has shed light on the intricate relationships between the ocular surface and its microbial inhabitants, including commensal bacteria and saprophytes, and how these interactions contribute to ocular health. This discussion delves into the composition of the ocular microbiome, the distinct roles played by commensal bacteria and saprophytes, and the impact of ocular antibiotics on this delicate microbial community.

The ocular microbiome comprises a variety of microorganisms, including bacteria, fungi, and viruses. Among these, bacteria are the most extensively studied component, represented by species belonging to the genera *Staphylococcus*, *Streptococcus*, *Corynebacterium*, and *Propionibacterium*, which are commonly found on the ocular surface. Additionally, fungi such as *Candida* and viruses like herpes simplex virus have been identified, though they have been studied less frequently.

More than 50 genera and several species have been identified via conventional culture and 16S rRNA analysis [18]. Maintaining microbial diversity (defined as the range of different kinds of unicellular organisms, bacteria, archaea, protists, and fungi) on the ocular surface is crucial to maintaining healthy and balanced ocular homeostasis [19].

Commensal bacteria and saprophytes are two key categories of microorganisms within the ocular microbiome, each playing a distinct role in maintaining ocular homeostasis.

Commensal bacteria are non-pathogenic microorganisms that coexist with their host without causing harm. On the ocular surface, commensal bacteria contribute to ocular health by promoting immune tolerance, preventing the overgrowth of potentially pathogenic bacteria, and modulating the local immune response. These microorganisms represent a crucial component of the ocular ecosystem, creating a balanced environment that supports overall ocular health.

Saprophytic bacteria are environmental microorganisms that derive nutrients from decaying organic matter. In the context of the ocular microbiome, saprophytes contribute to the competitive exclusion of pathogens by occupying niches and utilizing available resources. The presence of saprophytic bacteria on the ocular surface is thought to stimulate the local immune system, fostering a state of equilibrium that prevents colonization by harmful microbes. They compose the core microbiota, which is an excellent protective tool for the eye and which we should try not to destroy unnecessarily with the use of aggressive ocular preservatives or antibiotics [20,21].

When there are predisposing factors (disruptions to the ocular barrier, ocular lesions or corneal lacerations, or systemic and local comorbidity), these bacterial groups can occasionally transform into ocular pathogens.

While they are essential for treating ocular infections, the use of ocular antibiotics can have repercussions on the delicate balance of the ocular microbiome. By their nature, antibiotics target a broad spectrum of bacteria, affecting both pathogenic and commensal species. Thus, the prolonged or frequent use of antibiotics may disrupt the equilibrium of the ocular microbiome, potentially leading to dysbiosis.

The impact of antibiotics on commensal bacteria is of particular concern as alterations in their amount and diversity may compromise the protective functions they provide to the ocular surface, even after the healing of an infection. This disruption may result in increased susceptibility, the recurrence of infections, or colonization by opportunistic pathogens [22].

Understanding the effects of different antibiotics on the ocular microbiome is crucial for optimizing therapeutic strategies that minimize collateral damage to commensal bacteria and preserve ocular homeostasis. From this perspective, it seems reasonable that the adoption of a short antibiotic dose regimen could be considered and preferred.

## 4. Optimizing Ocular Infection Treatment: Preserving the Ocular Microbiota and Minimizing Disruption

Ocular infections demand a targeted and effective approach to ensure resolution while minimizing collateral damage to the delicate ocular microbiota. In recent years, there has been a growing emphasis on treatment strategies that consider preservative-free antibiotic formulations with infrequent dosages to better preserve the ocular microbiota [23,24].

The lack of preservatives in these ophthalmic drugs may avoid a negative impact on the ocular microbiota. In fact, chemical antiseptic compounds such as benzalkonium chloride (BAK) can disrupt the ocular microbiota due to their antimicrobial properties, which induce cytotoxicity, adverse effects (i.e., allergy), and tear film alterations. Thus, the availability of BAK-free or preservative-free eye drops is desirable [25]. Preservative-free antibiotic formulations are generally available as unit-dose eye drops. Indeed, moxifloxacin, as it is a last-generation and potent quinolone antibiotic, is the first and currently the only self-preserving multidose formulation approved by the European Pharmacopoeia according to its requirements for preservative effectiveness [26].

The antibiotic dosage regimen (ADR), or the duration of treatment and the concentration of the dose, is another important component to consider. Choosing antibiotics with appropriate dose regimens is a crucial aspect of preserving the ocular microbiota. The frequent and prolonged use of antibiotics can disrupt the balance of the ocular microbial community, leading to dysbiosis and potential complications. Opting for antibiotics with sustained-release formulations or those requiring less frequent administration helps achieve effective therapeutic levels while minimizing drug exposure. The selection of an antibiotic should also consider its spectrum of activity, aiming for an agent that specifically targets the causative pathogens without adversely affecting commensal bacteria. This targeted approach supports the resolution of the infection while mitigating the risk of antibiotic-induced dysbiosis [27].

Shortening the duration of antibiotic treatment is a key strategy to prevent the prolonged disruption of the ocular microbiota. An unnecessarily prolonged treatment can contribute to the development of antibiotic resistance, and it may negatively impact the diversity and number of commensal microorganisms on the ocular surface. Tailoring treatment duration to the specific infection and monitoring the patient’s response can help strike a balance between effective resolution and microbiota preservation.

Again, because of its pharmacological (pharmacodynamic and pharmacokinetic) features, moxifloxacin assures prolonged and effective concentrations on the conjunctiva with a few drops, avoiding the need for frequent doses. This implies that only a thrice-daily instillation for the whole therapy is effective, which is less than the usual topical quinolone regimen for the first days of therapy (up to eight instillations per day) [4,28]. Moreover, a reduced drug regimen may improve patient compliance, reducing the treatment failure rate [26].

Finally, the preservation of the ocular microbiota by using the best and least harmful practices must be mandatory. The best practice consists of a comprehensive approach that includes the judicious use of preservative-free formulations and the choice of effective antibiotics which require a short duration of therapy and an infrequent dose regimen. By adhering to these principles, healthcare practitioners can optimize therapeutic outcomes while mitigating the risk of dysbiosis and long-term ocular complications.

As mentioned, some antibiotics can have a deep impact on the ocular microbiota and diversity. Recently, it was demonstrated that species diversity, community structure, and the composition of the ocular surface microbiota were significantly changed after exposure to ceftazidime or tobramicyn and vancomycin; however, they tended to be restored within weeks after discontinuing antibiotic treatment [29]. This aspect is managed well by other antibiotics if the right dosage of antibiotic is maintained and its administration is not prolonged beyond what is necessary. The ocular surface comes into daily contact with microbes coming from the nose and mouth, as well as from the environment in which we live, so it is conceivable that the ocular flora can be replaced after a few weeks. However, it is important that the antibiotic used does not select resistant strains: using antibiotics with high bactericidal power and which act quickly will ensure that the bacteria on the ocular surface will have no chance of becoming resistant (see below).

These findings should guide future strategies for the rational administration of antibiotics in ophthalmology and should provide a basis for understanding the role of ocular bacterial flora in eye diseases.

## 5. Understanding Eye Infections and Bacterial Conjunctivitis: The Necessity of Timely Treatment for Preventive and Therapeutic Outcomes

Ocular infections, particularly bacterial conjunctivitis, are common ocular pathologies that, despite their generally benign course, warrant timely and appropriate treatment. Recognizing the conditions under which these infections occur is crucial, not only for addressing immediate symptoms but also for preventing potential complications, minimizing the risk of contralateral eye infection, and understanding the epidemiological aspects, especially in vulnerable populations such as children and individuals at higher risk [30].

Eye infections, including bacterial conjunctivitis, can occur when pathogens gain access to the ocular surface. This can happen through direct contact with contaminated surfaces, such as hands or shared items, or as a secondary infection following a viral upper respiratory tract infection. Bacterial conjunctivitis, characterized by redness, discharge, and discomfort, is often caused by bacteria like *Staphylococcus aureus*, *Streptococcus pneumoniae*, or *Haemophilus influenzae*.

A correct diagnosis and prompt treatment when symptoms arise are essential to prevent the spread of the infection and mitigate potential complications. Additionally, eye infections may result from a compromised ocular surface, such as in the case of contact lens wearers, individuals with pre-existing ocular conditions, or those with weakened immune systems [30,31].

Treating eye infections, even when they have a generally benign course, aims to achieve several important purposes. First, timely intervention helps alleviate symptoms and discomfort, improving the quality of life for affected individuals. Secondly, treatment is essential to prevent the spread of the infection to the contralateral eye. Many cases of bacterial conjunctivitis are initially unilateral, and prompt treatment can reduce the risk of bilateral involvement [32].

Moreover, properly treating eye infections helps prevent complications such as corneal involvement or secondary bacterial infections. Untreated or inadequately managed infections may lead to more severe consequences, emphasizing the importance of early and appropriate therapeutic measures.

Eye infections, especially bacterial conjunctivitis, exhibit variable incidence across different demographics. Children, due to their close contact in school and daycare settings, are particularly prone to developing eye infections. The higher likelihood of touching the face and eyes, coupled with less-developed hygiene practices, contributes to increased vulnerability in this population [33,34].

Vulnerable individuals, including the elderly and immunocompromised individuals, face an elevated risk of ocular infections due to weakened immune responses. Chronic medical conditions, medications, or invasive ocular procedures can further increase susceptibility to infections.

Understanding the epidemiology of ocular infections in these populations is vital for implementing targeted preventive measures, enhancing public health strategies, and ensuring timely interventions to minimize the impact of eye infections.

## 6. Antibiotics in Ophthalmology: A Spectrum of Efficacy and Application

Epidemiological considerations, particularly in children and vulnerable populations, highlight the need for tailored preventive measures and emphasize the importance of comprehensive eye care in vulnerable groups. Therefore, the correct use of antibiotics based on microbial epidemiology is a cornerstone that should be safeguarded and developed further.

The armamentarium of antibiotics available for ophthalmic use includes different classes of drugs, each with unique characteristics that influence their efficacy and safety profiles. The most important classes utilized in ophthalmology are chloramphenicol, aminoglycosides, and quinolones. Contrary to the traditional notion that antibiotics with potent efficacy and significant tissue penetration such as quinolones should be reserved exclusively for severe cases like keratitis, evolving perspectives promote their broader utility as first-line therapies, challenging the wrong perception that they are overly potent and should be reserved only for difficult infections [35]. In fact, this short-sighted strategy could result in high treatment failure rates and further costs due to the additional healthcare resources required to resolve the pathologies. The concept of “potency” for an ophthalmic antibiotic implies its efficacy at lower concentrations and under reduced regimens, a wide spectrum of action, and greater spreading and penetration in ocular tissues, which can prevent the occurrence of complications and assure healing in a short time. The employment of a potent, broad-spectrum, last-generation quinolone as a first choice rather than a last choice may determine faster symptom relief and a low rate of treatment failure, also due to the minimal microbial resistance rate of moxifloxacin eyedrops [36].

As a matter of fact, quinolones are useful and suitable for the treatment of ocular infections. As the field continues to evolve, the judicious selection of antibiotics based on each specific clinical scenario and microbial profile remains important for optimizing patient outcomes [37]. Moreover, quinolones’ safety profile and low incidence of adverse effects contribute to their widespread acceptance in ophthalmology [38].

Chloramphenicol is a bacteriostatic antibiotic which has been considered in ophthalmic practice for its spectrum activity limited to some Gram-positive and Gram-negative bacteria. It inhibits protein synthesis by binding to the bacterial 50S ribosomal subunit. Aminoglycosides, such as gentamicin and tobramycin, are bactericidal antibiotics with a focus on Gram-negative bacteria. They act by binding to the bacterial 30S ribosomal subunit, disrupting protein synthesis. They are commonly used in ophthalmology for conditions like bacterial keratitis, particularly when Gram-negative pathogens are implicated. The high incidence of microbial resistance and the risk of toxicity, including nephrotoxicity and ototoxicity, necessitate their judicious use and monitoring. Finally, quinolones, including levofloxacin and moxifloxacin, have become mainstays in ophthalmic practice due to their broad-spectrum activity and favorable safety profiles. They inhibit bacterial DNA synthesis by targeting DNA gyrase and topoisomerase IV.

As assessed before, re-addressing the notion that quinolones are too potent and should be reserved for difficult cases, their safety and efficacy profiles have allowed for their expanded use in various ophthalmic scenarios. Quinolones have been increasingly employed not only in the treatment of common conjunctivitis but also in prophylactic measures during ocular surgeries, highlighting their versatility and acceptance within the ophthalmic community.

Recent studies evaluated minimal inhibitory concentrations (MICs) for second- (ciprofloxacin and ofloxacin), third- (levofloxacin), and fourth-generation (moxifloxacin and gatifloxacin) fluoroquinolones against endophthalmitis and keratitis isolates. Compared with the second- and third-generation agents, both moxifloxacin and gatifloxacin had optimal MIC activity against fluoroquinolone-susceptible and fluoroquinolone-resistant Gram-positive bacterial strains isolated from endophthalmitis cases. Ciprofloxacin and levofloxacin were equally potent against Gram-positive bacteria. In addition, moxifloxacin was more potent than gatifloxacin against fluoroquinolone-resistant Staphylococci and Streptococci [39,40,41,42].

In a time–killing in vitro study, moxifloxacin killed *S. pneumoniae* and *H. influenzae* faster than tobramycin and gentamicin. Moxifloxacin achieved a 99.9% kill rate (3-log reduction) after approximately 2 h, suggesting its potential clinical benefit as a first-line treatment for bacterial ocular infections [43].

## 7. Antibiotic Resistance in Ophthalmology: Implications and Challenges

Antibiotic resistance is a global health concern involving many medical specialties, and ophthalmology is not an exception. In eye care, antibiotic resistance represents the diminished effectiveness of antibiotics in fighting bacterial infections due to the development of resistance mechanisms by bacteria. This phenomenon arises from the selective pressure exerted by the overuse or misuse of antibiotics, leading to the survival and proliferation of antibiotic-resistant strains [44].

In everyday ophthalmology practice, antibiotic resistance generates significant challenges. Common ocular infections, such as bacterial conjunctivitis, keratitis, and endophthalmitis, are often treated with topical or systemic antibiotics. However, the increasing prevalence of antibiotic-resistant strains compromises the efficacy of standard treatment regimens. This not only leads to prolonged and more severe infections but also heightens the risk of complications and vision-threatening outcomes [45].

Furthermore, the limited arsenal of effective antibiotics against resistant strains raises concerns about the potential for treatment failures. In the context of ophthalmic surgeries, such as cataract or refractive procedures, for which prophylactic antibiotic use is routine, the emergence of resistant bacteria could affect preventive measures, increasing the likelihood of postoperative infections.

The increasing resistance to aminoglycosides (tobramycin and gentamicin) [43] and older-generation quinolone antibiotics has evoked great attention regarding a judicious approach to antibiotic prescription. Proper diagnosis, targeted treatment, and adherence to recommended guidelines can help mitigate the development of resistance. Additionally, ongoing research into novel antimicrobial agents and alternative treatment modalities is imperative to counteract antibiotic-resistant ocular infections effectively.

Resistant *S. epidermidis* and *S. aureus* strains are also worth noting [46]. These bacteria are often responsible for infections, and the most common examples include conjunctivitis, blepharitis, corneal ulcers, and endophthalmitis. Most feared ocular infectious events are related to the use of contact lenses (in the case of conjunctivitis and corneal ulcers) or to ocular surgery (endophthalmitis). High incidences of ocular infection by *S. epidermidis* have been reported, and in some cases, they are superior to those achieved by *S. aureus* [47,48].

Microbial resistance to moxifloxacin, unlike ofloxacin and ciprofloxacin, should occur at very minimal rates because it requires the occurrence of a double mutation on both bacterial enzymes and the DNA gyrase and topoisomerase [49].

It is certain that there is no antibiotic today that does not induce resistance, including moxifloxacin. However, some characteristics, already partly mentioned above, such as tissue concentrations far higher than the MIC, the MSW (mutant selective window), a very low MPC (mutant prevention concentration), and the double target of action on two different enzymes of bacterial DNA, make the double mutation of bacterial DNA unlikely. Additionally, the inhibition of bacterial cell efflux pumps makes moxifloxacin ideal from ecological and resistance transmission points of view [26].

Indeed, in recent years, quinolones have been developed with special attention paid to their increased spectrum of action and their ability to penetrate various tissues, obviously without neglecting their lower ability to induce resistance over time (Figure 1).

## 8. Infection Prophylaxis in Ophthalmology: Rational Approaches and Evolving Perspectives

In ophthalmology, infection prophylaxis is a crucial aspect of patient care, especially in the context of ocular surgeries and procedures. However, the appropriateness and necessity of antibiotic prophylaxis are topics to be addressed considering new knowledge and emerging challenges. Understanding the circumstances in which prophylaxis is beneficial and incorporating contemporary insights into antimicrobial stewardship are essential aspects of optimizing patient outcomes [50].

Prophylactic antibiotic use in ophthalmology is traditionally associated with surgical interventions to prevent postoperative infections. Cataract surgery, the most frequently performed ophthalmic surgery all over the world, often involves the perioperative use of topical antibiotics to reduce the risk of endophthalmitis. Similarly, intracameral antibiotics have gained popularity as prophylactic measures during cataract surgery. However, the widespread adoption of prophylactic antibiotics has raised concerns about antibiotic resistance, emphasizing the need for a balanced and evidence-based approach [51].

Recent studies suggest that the routine use of prophylactic antibiotics may not be warranted in all cases. Advances in surgical techniques, improvements in sterile operating conditions, and the potential risks associated with antibiotic use have prompted a reevaluation of prophylactic strategies. As a result, individualized approaches based on patient characteristics, surgical complexity, and the prevalence infection rates are gaining prominence [52].

In an era of evolving knowledge, the proper use of antibiotics in ophthalmology is becoming increasingly critical. Antimicrobial stewardship programs aim to optimize the use of antibiotics, preserving their effectiveness and minimizing the risk of resistance. This includes reconsidering the necessity of prophylactic antibiotics in low-risk cases and exploring alternative strategies, such as enhanced surgical techniques and improved preoperative and postoperative care.

The European Society of Cataract and Refractive Surgeons (ESCRS) Endophthalmitis Study Group published a landmark study in 2007 demonstrating that the risk of infectious postoperative endophthalmitis was reduced nearly fivefold (0.34% to 0.07%) using perioperative intracameral cefuroxime [53]. Over the past decade, several intracameral or intravitreal preparations of antibiotics compounded with steroids have been developed. The recent advent of intravitreal triamcinolone acetonide–moxifloxacin (Tri-Moxi), composed of 1.5% triamcinolone acetonide and 0.1% moxifloxacin, offers one option for endophthalmitis prophylaxis. Recent studies have demonstrated that triamcinolone acetonide–moxifloxacin is an effective tool to control intraocular inflammation after cataract surgery compared to a standard topical eye drop regimen [54].

Other promising features are the ocular tissue concentrations achieved after the topical administration of moxifloxacin 0.5%, making it a good candidate not only for treatment but also for prophylaxis. A study reports the pharmacokinetic properties of besifloxacin, gatifloxacin, and moxifloxacin in the conjunctival tissue of healthy volunteers after their topical application [28]. The results show that all three fluoroquinolones were well tolerated and reached levels in the conjunctiva above the MIC90s of methicillin-sensitive *S. aureus* and *S. epidermidis* for at least 2 h.

A very recent clinical trial retrospectively examined more than 2,000,000 patients undergoing cataract surgery in uncomplicated and complicated eyes, both with and without intracameral moxifloxacin 0.5% as prophylaxis [55]. Moxifloxacin reduced the overall endophthalmitis rate, with phacoemulsification, with manual small-incision cataract surgery, and in eyes with posterior capsule rupture or requiring secondary surgery.

In conclusion, infection prophylaxis in ophthalmology requires a differentiated and evidence-based approach. The evaluation of territorial antibiotic resistance, the application of individualized strategies, adherence to evolving guidelines, and the appropriate choice of surgery for each patient will contribute to optimizing patient care and minimizing the impact of antibiotic resistance in ophthalmology.

## 9. Pharmacokinetic Properties of Moxifloxacin 0.5% Ocular Drops: A Comprehensive Overview

Moxifloxacin 0.5% ophthalmic solution is specifically designed for topical use in the eye, demonstrating a range of pharmacokinetic properties that sustain its efficacy in treating ocular infections. Understanding these properties (Table 1) is essential for optimizing therapeutic outcomes and ensuring the safety of this ophthalmic formulation.

Pharmacodynamics describes the relationship between the concentration of a drug over time at the site of infection and the pharmacologic and toxicologic effects of the drug. It depends on the mechanism of action against the bacteria (i.e., bacterial inhibition or killing) and can be described as either time- or concentration-dependent. For these agents, such as fluoroquinolones, the higher the drug concentration (i.e., above the MIC), the more rapid and wider the bacterial eradication, with little chance of generating resistant strains. The goal of the dose regimen with these antibiotics is to maximize the drug concentration in the ocular tissues. In these circumstances, ophthalmic products with higher concentrations of antibiotics offer better assurance that the antibiotic can be present at or above the MIC for a longer time [56].

Therapeutic efficacy depends on the concentration of antibiotics at the target site, primarily the concentration of free, or unbound, drug. Most fluoroquinolones rapidly penetrate ocular tissues, achieving tissue concentrations that are generally higher than those found in plasma.

Moxifloxacin demonstrated higher concentrations in the tear film and greater penetration into anterior ocular tissues than older fluoroquinolones because it is more soluble at the normal physiologic pH of 7.0 at the ocular surface and because it is both lipophilic and hydrophilic [57].

Moxifloxacin is formulated at a pH of 6.8, while gatifloxacin is formulated at a pH of 6.0 [4]. In addition, moxifloxacin is available at a higher concentration than gatifloxacin (0.5% vs. 0.3%) [58], and moxifloxacin achieves greater tissue penetration than gatifloxacin. Data evaluated in intact rabbit corneas suggest differences in the ocular penetration of these fluoroquinolones: a solution of moxifloxacin at 0.3% (i.e., 40% lower or 0.2% less than the commercial formulation) achieved high concentrations in rabbit ocular tissues within 30 min after a single dose (12.5 µg/mL in the cornea and 1.8 µg/mL in the aqueous humor), which was higher than gatifloxacin 0.3% (4.5 µg/mL in the cornea and 0.27 µg/mL in the aqueous humor 1 h after administration). Moxifloxacin reached almost a threefold higher concentration in the cornea compared with gatifloxacin in approximately half the time. This has very important pharmacodynamic significance as the concentrations and the MICs for susceptible isolates reached by moxifloxacin are much higher among all quinolones.

A recent open-label human pharmacokinetic study measured moxifloxacin’s penetration into the aqueous humor in human adults undergoing cataract surgery. The Cmax for 21 patients who received topical moxifloxacin preoperatively exceeded the MIC values for the main strains responsible for endophthalmitis, such as *S. aureus* and *S. epidermidis*. The attained concentrations significantly exceeded the MIC90 (the minimum inhibitory concentration of an antibiotic required to inhibit the growth of 90% of bacteria strains) levels for most organisms causing ocular infections. The aqueous penetration of moxifloxacin in both groups was not significantly affected by gender, intraocular pressure, or comorbidities [59].

Another study [42] reported an MIC90 value of 0.063 µg/mL against clinical isolates of methicillin-susceptible *S. aureus*. The therapeutic index (which combines in vitro MIC data with in vivo penetration data to compare the clinical efficacy of a certain antibiotic) for moxifloxacin using these values would be 29.2. This means that moxifloxacin delivers substantially more drug than is necessary to inhibit bacterial growth, but it also delivers substantially more drug than required to prevent the selection of resistance.

Wico W. et al. [60] demonstrated the good penetration of moxifloxacin into the aqueous and vitreous humors after administration before vitrectomy. The aqueous (1.576 ± 0.745 µg/mL) and vitreous (0.225 ± 0.013 µg/mL) levels of moxifloxacin were very high and far exceed the MIC90s for a wide variety of bacteria implicated in endophthalmitis.

Following a single dose of moxifloxacin in 15 patients, the conjunctival levels of moxifloxacin were statistically significantly higher than those of the other four fluoroquinolones (*p* < 001). The mean conjunctival concentration of moxifloxacin (18.0 µg/g) was from 6.8- to 7.7-fold higher than the others [61].

These data corroborate some studies that were also conducted in children. To assess the efficacy of topical antibiotic therapy for acute infective conjunctivitis in children, a randomized clinical trial conducted in a primary healthcare setting demonstrated that topical antibiotic treatment with moxifloxacin was associated with significantly shorter durations of conjunctival symptoms (3.8 vs. 5.7 days; difference, −1.9 days; 95% CI, −3.7 to −0.1 days; *p* = 0.04) [9].

Interestingly, a review of five independent, multicentered, double-masked, parallel, controlled studies was conducted to determine the safety and efficacy of moxifloxacin ophthalmic solution 0.5% in pediatric and nonpediatric patients with bacterial conjunctivitis [10]. Based upon this review, moxifloxacin ophthalmic solution 0.5% formulated without the preservative benzalkonium chloride is safe and well tolerated in pediatric (3 days–17 years of age) and nonpediatric (18–93 years) patients with bacterial conjunctivitis.

## 10. Conclusions

Moxifloxacin 0.5% ophthalmic eye drops stand out as a milestone in the kingdom of ocular therapeutics, offering a delicate balance between efficacy, tolerability in maintaining ocular homeostasis, and optimal pharmacokinetic characteristics. The multifaceted attribute of moxifloxacin makes it as a flexible and reliable choice in the treatment of ocular infections.

One of the pivotal aspects contributing to the favorable role of moxifloxacin in ocular homeostasis is its safety profile, especially in children. The minimal systemic absorption of moxifloxacin from topical ocular administration minimizes the risk of systemic side effects, aligning with the principles of antimicrobial stewardship. This feature is crucial in preserving the delicate ocular microbiota, preventing dysbiosis, and supporting the overall health of the ocular surface.

Regarding its impact on the ocular microbiota, correlation studies between moxifloxacin and this important ocular “organ” should certainly be implemented. A study using a traditional molecular assay aimed to evaluate whether moxifloxacin 0.5% ocular drops and wearing soft contact lenses predispose patients to pathogen colonization and/or infection in a very small cohort of 10 patients. Despite the prophylactic use of moxifloxacin, the eye surface was not completely sterile in some patients (samples from the inferior conjunctival fornix and the KPro–donor cornea interface were positive for coagulase-negative staphylococcus in three patients and for *Aerobasidium pullulans* in one patient). None of the patients with culture-positive results developed keratitis or endophthalmitis during the study [62]. Obviously, this study is too small to draw conclusions on the impact that moxifloxacin can have on the ocular microbiota, but it demonstrates how the ocular surface and its appendages can be easily colonized by resident or transient bacteria.

In cases of infection, moxifloxacin’s broad spectrum of activity against both Gram-positive and Gram-negative bacteria further enhances its therapeutic utility and also avoids the employment of preservatives in multidose containers, preserving the ocular surface and microbiota. Its efficacy in addressing a diverse range of ocular infections, including bacterial conjunctivitis, keratitis, and postoperative prophylaxis, highlights its versatility in clinical practice. This broad spectrum allows for a targeted approach, addressing the specific microbial etiology of each case while minimizing the risk of resistance development.

The optimal pharmacokinetic characteristics of moxifloxacin, including rapid absorption, widespread distribution within ocular tissues, and sustained therapeutic levels, contribute to its efficacy and benefits. These attributes enable less-frequent dosing while ensuring a prolonged duration of action, a valuable feature in enhancing patient compliance and facilitating effective treatment regimens.

Looking toward the future, moxifloxacin continues to play a pivotal role in ocular therapeutics. Ongoing research and clinical experience may further refine our understanding of its applications, potentially expanding its use in emerging challenges such as antibiotic-resistant strains. Additionally, as the field of ophthalmology advances, the proper use of moxifloxacin, informed by evolving knowledge and evidence-based practices, will remain essential in optimizing patient outcomes.

In conclusion, moxifloxacin 0.5% ophthalmic drops emerge not only as a potent antimicrobial agent but also as a guardian of ocular homeostasis. Moxifloxacin’s favorable safety profile, broad-spectrum activity, and optimal pharmacokinetics make this molecule a reliable choice for clinicians, ensuring effective treatment while preserving the delicate balance of the ocular microbiota.

## Figures and Tables

**Figure 1 microorganisms-12-00649-f001:**
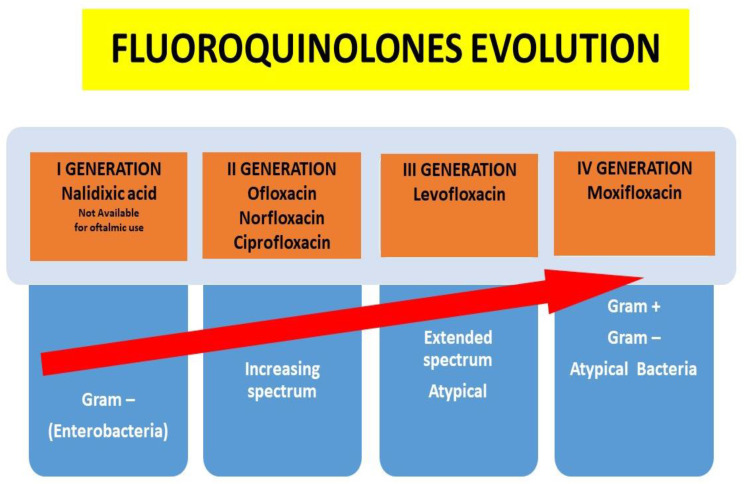
Quinolones evolution according to the increased spectrum of activity and pharmacokinetic characteristics.

**Table 1 microorganisms-12-00649-t001:** Pharmacokinetic features of an ideal topical antibiotic.

Parameters	Meaning
Peak concentration achieved in the tissue (Cmax)	Concentrations must be always above the MICs
Time to maximum concentration in the tissue	Achieve quickly the right concentration for a rapid bacterial killing and avoiding resistance
Time of Elimination	Rapid amount elimination avoids toxicity and sub-inhibitory concentrations
Half-life	Necessary to calculate the administration time over the day
Penetration at different sites	Tissue and compartment penetration avoid infection spreading

## Data Availability

The data that support the findings of this study are openly available in pubmed and google scholar. The author confirm that the data supporting the findings of this study are available within the articles cited in the text (References).

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
