# Peer review of "Topical Antibiotic Therapy in the Ocular Environment: The Benefits of Using Moxifloxacin Eyedrops"

_microorganisms, 2024, doi:10.3390/microorganisms12040649_

Round 1

Reviewer 1 Report

Comments and Suggestions for Authors

The authors have written a comprehensive review about the use of Moxifloxacin in ocular diseases. The article is clear and logically structured, offering a smooth flow from Moxifloxacin's pharmacological attributes to its clinical implications. The academic tone is maintained throughout, with appropriate technical language and citations.

The article provides a thorough overview of Moxifloxacin's mechanism, efficacy, and pharmacokinetics, particularly in the context of ocular applications.

It emphasizes the importance of considering the ocular microbiota and microbial resistance when choosing an antibiotic for eye infections, which is crucial for long-term ocular health. The discussion on the clinical applications of Moxifloxacin in treating various ocular conditions and its safety profile makes the review highly relevant for practitioners in ophthalmology.

While the article mentions microbial resistance as an uncommon event for Moxifloxacin, it lacks a detailed analysis of potential resistance mechanisms and their implications.

There's a need for more information on the long-term impact of Moxifloxacin usage on ocular microbiota and resistance patterns, which would be critical for its sustained use.

Author Response

Dear Reviewer, I would like firstly thank for the positive comments regarding the article. As suggested, I have further clarified some points about the resistance induction mechanisms of moxifloxacin, emphasizing its  appropriate use by respecting dosage and time of treatment (and avoiding its long term useage).

Regarding the microbiota, the related papers are few and not very clear (I have now included a further study in the discussion), for this reason I also emphasized in the discussion the future need to clarify the impact of moxifloxacin on the ocular microbiota using better methods (NGS) and strategies.

Reviewer 2 Report

Comments and Suggestions for Authors

I found this review to be interesting and very informative.

I have a few comments.

This is quite a comprehensive review and not solely about moxifloxacin. I would suggest that the title is changed to reflect this - to something like "Chemotherapy in the ocular environment; the benefits to the use of moxifloxacin" or something like this. I would suggest that the conclusion is also altered appropriately.

In a few places I found the English to be less than perfect and would recommend that a native English speaker reads the manuscript and adjusts accordingly.

The authors has two addresses but one number to his name.

Comments on the Quality of English Language

Correct as suggested above

Author Response

Dear Reviewer, very appreciated the positive comments, thank you very much.

As suggested, I have considered the recommended title. Now it is changed accordingly.

The English is now reviewed by a native language person and the entire manuscript checked for that.

I have also added the number to my second affiliation.

Reviewer 3 Report

Comments and Suggestions for Authors

The paper by Lorenzo Drago is an attempt to review the advantage of Moxifloxacin for the oftalmology. While the problems raised by author are of high importance, the paper looks loke an advertisment of this antimicrobial and therefore is not acceptable at present form and should be re-written. 

Abstract shoul be re-written in direction to remove all discussions and stress on facts, advantages and side effects of moxi.

Paper lacks of data showing changes in eye micorbiota, resistance develpment etc

Only partially the efficiency of the drug against various pathogens is described, like a scatter in the text. 

I would suggest to focus on following issues:

- molecular mechanism af moxi activity and problems of bacterial susceptibility to the drug,

- resistance to moxi and its development, mechanisms, risks etc,

- special features of antimicrobial treatment of eyes, pharmacokinetics of drug diffusion  etc

- effect of treatmnet on eye microbiota. 

Author Response

I would like to thank the Reviewer for the many comments and suggestions, which in truth have already been partly discussed and dissected in the manuscript.

As suggested, the abstract has been revised in some parts and has covered the pros and cons of moxifloxacin (including the lack of data on the the ocular microbiota).

Regarding the spectrum of activity of moxifloxacin, it has been only mentioned along the text, as it has been already matter of many other reviews and, considering the length of the manuscript, I did not wish to repeat what has already been widely stated by other authors.

Accordingly, I added in the manuscript the mechanisms of moxifloxacin in reducing the induction of resistance, while the kinetic and diffusion data of this drug have already been discussed in the manuscript (again I’ve tried to summarize these data avoiding to be redundant with what has already been highlighted in the literature).

Thanks for the comment on the ocular microbiota. Unfortunately, as is known, the literature on the impact of various antibiotics, including moxifloxacin, on the ocular microbiota is lacking. Now, I've added one more study into the discussion, but it is old and considers few patients. The data are scarce and discrepant from each other, so I have emphasized this both in the abstract and in the discussion as well as the needs to perform more studies by using NGS methodology and new clinical strategic approaches.

Round 2

Reviewer 3 Report

Comments and Suggestions for Authors

The paper has been improoved in part of the abstract and concluding remarks, and overall the text was corrected. May be, the information present here will be helpfull for clinicians